# Runs of Homozygosity Analysis Reveals Genomic Diversity and Population Structure of an Indigenous Cattle Breed in Southwest China

**DOI:** 10.3390/ani12233239

**Published:** 2022-11-22

**Authors:** Wei Wang, Yi Shi, Fang He, Donghui Fang, Jia Gan, Fuqiu Wu, Yueda AG, Xiaodong Deng, Qi Cao, Chu Duo, Wangdeng RZ, Maozhong Fu, Jun Yi

**Affiliations:** 1Animal Breeding and Genetics Key Laboratory of Sichuan Province, Sichuan Animal Science Academy, Chengdu 610066, China; 2Jiulong County Agriculture, Animal Husbandry and Science and Technology Bureau, Jiulong 626200, China

**Keywords:** Xieka cattle, ROH, SNPs, inbreeding, Sichuan

## Abstract

**Simple Summary:**

For the reason that the systematic breeding programs and pedigree records are unavailable in the indigenous livestock breeds, it is hard to evaluate their genetic diversity and population structures using the traditional pedigree-based and demographic approaches. The Xieka cattle are an indigenous breed geographically distributed in the southeastern Sichuan, China, and have the long-term evolutionary adaptation to local subtropical highland environments. To explore efficient programs on genetic resources conservation and utilization, the genetic diversity and population structures of Xieka cattle were investigated in this study using genomic information. Our analyses revealed that this indigenous cattle breed have remained a relatively high degree of genetic diversity and have not suffered from the recently generated inbreeding. Furthermore, some candidate genomic regions and genes were suggested to be likely associated with the diverse production traits in cattle.

**Abstract:**

In aiming to achieve sustainable development goals in the livestock industry, it is becoming increasingly necessary and important for the effective conservation of genetic resources. There are some indigenous cattle breeds in Sichuan, southwest China, for which, however, the genetic diversity and population structures still remain unknown because of the unavailability of systematic breeding programs and pedigree information. Xieka cattle are an indigenous breed locally distributed in southeastern Sichuan and have a long-term evolutionary adaptation to local environments and climates. In this study, we obtained 796,828 single nucleotide polymorphisms (SNPs) through sequencing the genomes of 30 Xieka cattle and used them for analyzing the genetic diversity and runs of homozygosity (ROH). The mean nucleotide diversity was 0.28 and 72% of SNPs were found to be in the heterozygous states. A total of 4377 ROH were detected with even distribution among all autosomes, and 74% of them were lower than 1 Mb in length. Meanwhile, only five ROH were found longer than 5 Mb. We further determined 19 significant genomic regions that were obviously enriched by ROH, in which 35 positional candidate genes were found. Some of these genes have been previously reported to be significantly associated with various production traits in cattle, such as meat quality, carcass performances, and diseases. In conclusion, the relatively high degree of genetic diversity of Xieka cattle was revealed using the genomic information, and the proposed candidate genes will help us optimize the breeding programs regarding this indigenous breed.

## 1. Introduction

Because of the relative low productivity on the economically important traits in context of modern livestock industry, indigenous breeds are facing significantly decreasing population sizes and increasing inbreeding worldwide [1]. However, these gene pools are valuable due to their long-term evolutionary adaptation to the diverse local environments. There have been abundant indigenous cattle genetic resources in Sichuan Province, China, with seven officially recognized breeds (*Bos taurus*), as well as some unregistered breeds and populations. Wang and colleagues (2018) [2] investigated the genetic diversity among six indigenous cattle breeds in Sichuan using genome-wide single nucleotide polymorphisms (SNPs) that were obtained from the restriction site-associated DNA sequencing (RADseq) approach. However, the genomic diversity and population structures have not been studied yet for these unrecognized cattle breeds officially in Sichuan. The Xieka cattle are an indigenous breed with small body size, which have the average adult live weights of 370 Kg and 280 Kg for males and females, respectively (according to our field investigation). It is estimated that the current population size of Xieka cattle is ~15,000 having been mostly distributed in Jiulong county, which is geographically located in southeastern Ganzi Tibetan Autonomous Prefecture, Sichuan Province, China.

Using the genome-wide short tandem-repeat polymorphisms, Broman and Weber (1999) [3] first found the prevalent occurrences of long homozygous chromosomal segments in humans, and these homozygous segments were termed as the runs of homozygosity (ROH) in the accompanying editorial comments [4]. However, the relevant studies of ROH had not been comprehensively carried out until the presences of high-throughput genotyping technologies, including the second-generation genome sequencing approaches and SNP assays [5]. For the reason that ROH are theoretically mainly derived from parental inbreeding, they have been extensively involved in studying both population structures and demographic history in human and livestock [6,7]. Notably, Curik et al. (2014) [8] reviewed the application of ROH analysis for estimating inbreeding at the individual and population levels and compared different relevant population statistics. The genome sequence variants and pedigree data in Holstein cattle were comprehensively evaluated for different measures of inbreeding [9]. In domestic cattle, genome-wide ROH was first investigated using 777,962 SNPs among nine worldwide modern breeds [10], which provided an excellent research framework in relation to ROH in livestock. Subsequently, the ROH analyses of cattle were often reported, for instance, in the U.S. Holstein cattle regarding the artificial selection signatures [11] and bull fertility [12], and in Polish and Chinese beef cattle associated with the beef production traits [13,14].

Aiming to understand the current genetic landscape of the gene pool of Xieka cattle, in this study, we employed the genome resequencing data for ROH analyses and further investigating the genomic diversity and population structure regarding this indigenous breed in China.

## 2. Materials and Methods

### 2.1. Ethics Statement

All blood samples involved in this study were collected by veterinarians at the annual health inspection, which means that no ethical approval is required.

### 2.2. Animals and Sample Collection

A total of 30 adult Xieka cattle were collected in this study, consisting of 12 males and 18 females. To guarantee our sampling as being representative as possible, we recruited the candidate animals according to the two considerations. First, these sampled animals were separately raised on rural farmers and did not have known pedigree relationships with each other. Second, individual morphological characteristics were carefully checked to avoid possible hybrid offspring most likely from the exotic cattle breeds. Blood samples from the external jugular vein were collected and used for the genome sequencing.

### 2.3. Genomic DNA and Sequencing

Genomic DNA was extracted using the Axy-Prep Genomic DNA Miniprep Kit (Axygen Bioscience, Tewksbury, MA, USA). The paired-end sequencing libraries with 350 bp in length were constructed according to Illumina’s protocol (Illumina, San Diego, CA, USA). In brief, 0.5 μg of genomic DNA was fragmented, end-paired, and ligated to adaptors, respectively. The ligated fragments were subsequently fractionated on the agarose gels and purified by PCR amplification to produce sequencing libraries. The successfully constructed libraries were sequenced on the Illumina HiSeq platform and the 150 bp paired-end reads were finally generated (Novogene Co., Ltd., Beijing, China).

### 2.4. SNP Genotyping and Quality Controls

The raw sequencing reads in FASTQ format were subjected to quality filtering to discard low-quality reads using the fastp software v0.23.2 [15], which were categorized into one of the following types: (i) reads contaminated by adaptor sequences, (ii) reads containing unambiguous bases of N more than 10% of total length, and (iii) reads containing low-quality bases (i.e., the reported Quality value of base <5) more than 50% of the total length. If any member of the paired reads was marked as low quality, both pairs were discarded. After these steps, we obtained clean reads and mapped them to cattle reference genome (ARS-UCD1.2) using BWA mapper v0.7.17 with the default parameters [16]. Subsequently, we employed GATK toolkit v4.2.5.0 [17] for the SNP discovery and genotyping across all samples according to the GATK Best Practices recommendations [18], in which the duplicate removal, InDel realignment and hard filtering algorithms were performed with the default parameters.

After obtaining raw SNPs, we first extracted these biallelic SNPs with Quality value > 20, coverage depth > 3, and being located on the 29 autosomes using VCFtools v0.1.16 [19]. Second, these SNPs were further subjected to population-based quality filtering using PLINK software v1.9 [20], as requiring the minor allele frequency (MAF) > 0.05, missing genotype rate < 0.1, and no significant deviation from Hard-Weinberg equilibrium (HWE, *p* > 10^−6^). Finally, the missing genotypes were also imputed using Beagle software v5.3 with the default parameters [21].

### 2.5. Genetic Diversity and Runs of Homozygosity

First, we investigated the genomic distribution regarding all clean SNPs and calculated three genetic diversity statistics, including the MAF, or nucleotide diversity on a per-site basis [$=\frac{n}{n-1}\left(1-\sum x_{i}^{2}\right)$, where n is the total number of sequences and $x_{i}$ is the observed frequency of i allele], and observed heterozygosity using VCFtools v0.1.16 [19]; meanwhile, we calculated the pairwise individual relatedness based on the method of Yang et al. (2010) [22]. Second, we detected the genomic ROH using the detectRUNS (v0.9.6) R package [23]. The individual homozygous segments were determined in every animal using a sliding window of 50 SNPs, in which no more than one heterozygous SNPs and five missing SNPs were allowed. Subsequently, an effective ROH was defined by requiring the minimum number of 100 SNPs contained, at least 500 Kb in length, the minimum SNP density of 1 SNP per 50 Kb, the minimum proportion of homozygous overlap window of 0.05, and the maximum gap between continuous homozygous SNPs, which was 100 Kb. The significant genomic regions were determined if they were simultaneously coveraged by ROH among more than 30% of all samples.

### 2.6. Functional Analysis

Within the significant genomic regions, the annotated functional genes were explored using biomaRt R package [24]. The ARS-UCD1.2 assembly was used as reference genome. Regarding these candidate genes, functional enrichment analyses were conducted using the DAVID web tool (accessed on June 15, 2022) [25] regarding the Gene Ontology (GO) terms and Kyoto Encyclopedia of Genes and Genomes (KEGG) pathways. The default Benjamini-Hochberg method was used for computing *p* values with the threshold of 0.05.

## 3. Results

### 3.1. SNPs and Genetic Diversity

We obtained a total of 1711 million raw paired sequencing reads, from which 1704 million clean reads (with the mean of 56.8 million per sample) were finally generated after quality filtering (Appendix A). Against the reference genome, average mapping rate was 99.5% for these clean reads; and, by which, the 96.9% and 57.2% of genome sequences were covered by at least 1X and 4X sequencing reads, respectively (Appendix A). A total of 48,699,312 raw SNPs were initially obtained, which resulted into 796,828 clean SNPs after being subjected to our custom processing steps. These high-quality SNPs were evenly distributed among the 29 autosomes (Figure 1A), with a transition/transversion ratio of 2.25.

Regarding the three statistics of genetic diversity that were calculated by SNPs, their statistical distributions are shown in Figure 1B. Among all SNPs, the mean and median of MAF were 0.19 and 0.15, respectively; meanwhile, the nucleotide diversity ranged from 0.09 to 0.51 (with the mean of 0.28). Furthermore, on average 72% (ranging from 68% to 76%) of SNPs were found to be in the heterozygous states among all the 30 samples. Among these animals, only a small proportion of pairwise comparisons showed the relatively high genetic relatedness (Figure 1C), and all of which had the mean and median of −0.03 and −0.05, respectively.

### 3.2. Genomic Patterns of Runs of Homozygosity

A total of 4377 ROH were detected among the 30 individuals, and all of them were evenly distributed across the 29 autosomes (Figure 1D). The highest and lowest numbers of ROH were observed in the chromosome BTA1 (*n* = 282) and BTA29 (*n* = 41), respectively. On the whole, there were 3226 ROH with length of 0.5–1 Mb (74%), 970 ROH of 1–2 Mb (22%), 176 ROH of 2–5 Mb (4%), and five ROH > 5 Mb. Within each individual, the length distributions of ROH are shown in Figure 2, which revealed the observable differences on their numbers and lengths of ROH; the mean (±standard deviation) of total ROH lengths was 132.9 ± 81.9 Mb, and whose proportions present in the genome (i.e., the ROH-based inbreeding coefficients) individually ranged from 1.6% to 16.7% (mean = 5.3%). Furthermore, the five longest ROH (>5 Mb) were found in five different animals (Figure 2).

There were 1263 SNPs being located within the ROH that were simultaneously found in more than 30% of all individuals. These SNPs were further clustered into 19 significant genomic regions that have been distributed among nine chromosomes, including the BTA2 (*n* = 2), BTA6 (*n* = 2), BTA9 (*n* = 2), BTA11 (*n* = 1), BTA15 (*n* = 1), BTA16 (*n* = 2), BTA17 (*n* = 1), BTA21 (*n* = 6), and BTA27 (*n* = 2), respectively (Table 1). Among them, the four longest genomic regions were found on both BTA15 (895 Kb in length) and BTA21 (703 Kb, 610 Kb, and 606 Kb in length).

### 3.3. Genes within the Significant Genomic Regions

Within these significant genomic regions, we found a total of 34 protein-coding and one microRNA genes (Table 1 and Appendix A), and these genes were mainly distributed on the three chromosomes of BTA27, BTA21, and BTA11, respectively. Based on the functional enrichment analyses regarding these candidate genes, four GO terms (GO:0005887~integral component of plasma membrane, GO:0009986~cell surface, GO:0043005~neuron projection, and GO:0005783~endoplasmic reticulum) and one KEGG pathway (bta04080:Neuroactive ligand-receptor interaction) were revealed.

## 4. Discussion

Jiulong county has subtropical highland climates with annual mean temperature of 9.1–17.5 °C [26]. Meanwhile, this region has been relatively isolated because of geography and poor transportation, especially in these rural areas. Accordingly, we believe these local genetic resources of domestic cattle, as well as other livestock, would be valuable for achieving sustainable development goals in future. The Xieka cattle are an indigenous breed but have not been officially registered yet. Understanding genetic diversity and population structures is necessary for establishing efficient breeding and conservation programs. Unfortunately, it is difficult or impossible in such an indigenous breed to obtain the accurate pedigree information, which has restricted the possible application of traditional approaches [27]. In this study, therefore, we obtained genome-wide SNPs in Xieka cattle and used them for evaluating genomic diversity and population structure. To the best of our knowledge, this is the first genetic study in Xieka cattle using the genomic information.

Using genome-wide SNPs generated from RADseq approach, Wang et al. (2018) [2] analyzed six officially registered indigenous cattle breeds in Sichuan and found their mean nucleotide diversity was 0.19 with the ranges from 0.26 in Ganzi cattle to 0.31 in Pingwu cattle. In this study, we similarly found the comparable nucleotide diversity with a mean of 0.28, and most SNPs were in the heterozygous states. Compared with previous reports of ROH in cattle [10,13], it seems that a smaller number of ROH were detected in this study; however, the total numbers and distribution of length sizes could not be directly compared due to differences on sample sizes and ROH definitions among the different studies. According to the total length of ROH, the estimated inbreeding coefficients in Xieka cattle (5.3%) are lower than the previous relevant estimates in other cattle breeds, such as in U.S. Holstein cattle [12], and in Hereford, Montbeliarde, and others [13]. These results indicate that Xieka cattle have retained the considerable genetic diversity despite its relatively small population size in comparison with the geographically adjacent cattle breeds. Furthermore, more than 70% of the detected ROH in Xieka cattle have length less than 1 Mb and only a few ROH were longer than 5 Mb, which suggests the low inbreeding levels and also the absence of recent inbreeding. Taken together, our genomic analyses confidently revealed that the Xieka cattle have not suffered from the serious losses of genetic diversity and obvious inbreeding.

Based on the ROH analysis, nine genomic regions were found to be significantly enriched in low-fertility when compared with high-fertility bulls [12]. In Chinese Simmental cattle, Zhao et al. (2021) [14] conducted association analyses between the enriched genomic regions of ROH and beef production traits and found many biologically meaningful candidate genes. In eight different livestock and pet species, Gorssen et al. (2021) [28] conducted a comprehensive ROH analysis and made the results publicly available for finding potential artificial or natural selection signatures and candidate functional genes. Therefore, ROH analyses are becoming a state-of-the-art approach for exploring the population structures and mapping the critical genomic regions and candidate genes associated with the economically important traits in cattle and other livestock species. In this study, we found some significant genomic regions, especially located on the BTA21 and BTA15, that were obviously enriched by ROH. Furthermore, four GO terms and one KEGG pathway were revealed to be associated with the candidate genes that are located in these significant genomic regions. Among them, the neuroactive ligand-receptor interaction pathway was recently reported to be significantly associated with heat stress in Australian Holsteins, which may participate in maintaining metabolic homeostasis in cattle during thermal stress [29].

We further conducted literature searches for investigating the possible biological functions of candidate genes revealed by ROH analysis in this study. Among them, the *CORIN* (corin serine pepsidase) gene, a member of the trypsin superfamily located on BTA6, was reported to be associated with meat colour traits in Nellore cattle [30]. The *SPAST* (spastin) gene was proposed to be the affected gene of spinal dysmielination disease in cattle [31]. In humans, the genetic variants of *TNFSF4* (TNF superfamily member 4) gene were significantly associated with the primary Sjögren’s syndrome [32]. In Original Braunvieh cattle, the *TMEM201* (transmembrane protein 201) gene was suggested to be related to selection signatures [33]. Buchanan et al. (2016) [34] analyzed the triacylglycerol and phospholipid fatty acid fractions in angus cattle and reported *INPP4B* (inositol polyphosphate-4-phosphatase type II B) as a candidate gene. Other functional genes revealed in this study were also previously reported to be associated with diverse production traits, including the *RCN2* (reticulocalbin 2), *PSTPIP1* (proline-serine-threonine phosphatase interacting protein 1), *ADRB3* (adrenoceptor beta 3) genes with carcass traits in Hanwoo and Qinchuan cattle [35,36], *GABRB3* (gamma-aminobutyric acid type A receptor subunit beta3) and *GABRB5* (gamma-aminobutyric acid type A receptor subunit alpha5) genes with temperament in beef cattle [37], *ZNF703* (zinc finger protein 703) and *ERLIN2* (ER lipid raft associated 2) genes with average daily gain in Nellore and with residual concentrate intake in Holstein [38], and *GOT1L1* (Glutamic-oxaloacetic transaminase 1 like 1) as a cell indicator of stress in the cattle semen [39]. Interestingly, we also mapped a microRNA gene of bta-mir-2400 that was experimentally revealed for affecting the preadipocytes proliferation and differentiation in yak [40], regulating skeletal muscle satellite cells proliferation in Chinese Simmental cattle [41]. On the whole, the current literature evidence confirms that the revealed significant genomic regions are likely associated with the local environmental adaptations and other production traits in Xieka cattle.

## 5. Conclusions

In this study, we employed the genomic information and ROH analyses for investigating genetic diversity and population structure regarding an indigenous cattle breed that are geographically distributed in Southwest China. Our results revealed the comparable genetic diversity to other geographically adjacent indigenous cattle breeds. Furthermore, we determined the significant genomic regions that are enriched by ROH, in which some known and novel genes were suggested to be associated with production traits in cattle.

## Figures and Tables

**Figure 1 animals-12-03239-f001:**
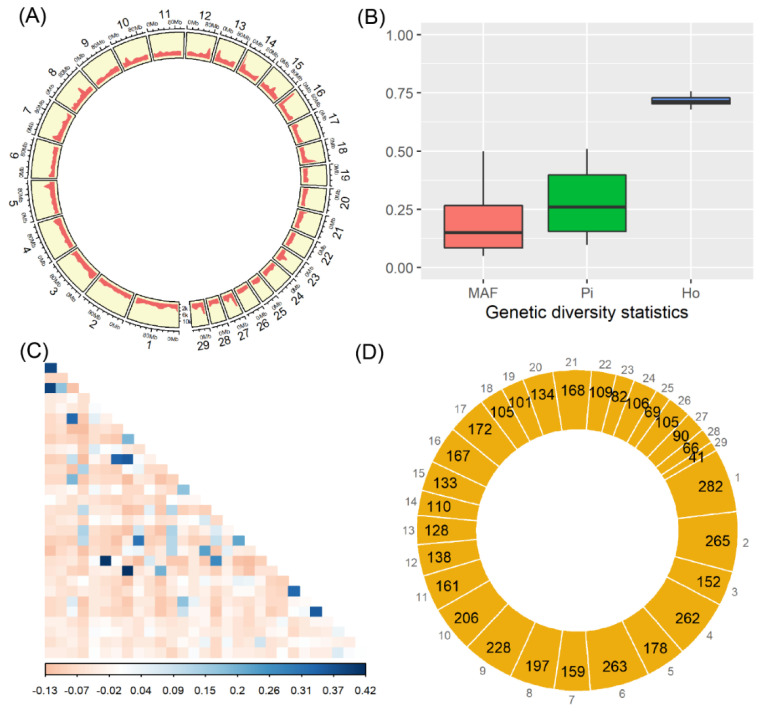
Results of SNPs, genetic diversity, and runs of homozygosity. All SNPs were illustrated for their genomic distribution, with the number of SNPs (y-axis) per 10 Mb genomic region (**A**), the calculated statistics of minor allele frequency (MAF), nucleotide diversity (Pi), and the observed heterozygosity (Ho) in (**B**), and the derived pairwise relatedness among individuals (**C**). Among the 29 autosomes, the numbers of runs of homozygosity are schematically shown in (**D**).

**Figure 2 animals-12-03239-f002:**
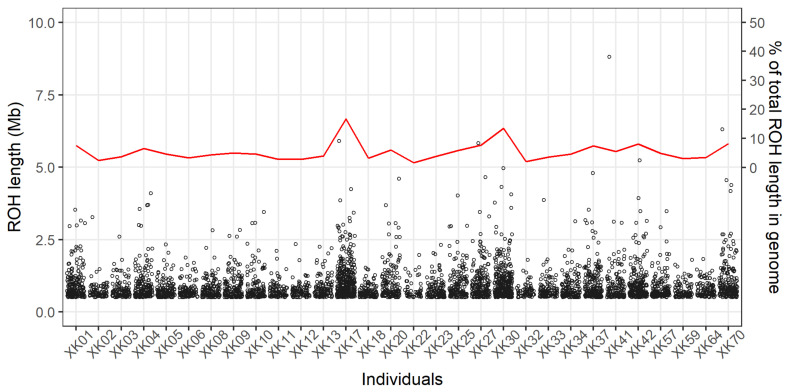
Numbers and lengths of runs of homozygosity. All runs of homozygosity are represented by the hollow points with their lengths showing on the left-hand y-axis. The individual genomic proportions (the red line) for total length of runs of homozygosity are shown in the right-hand y-axis.

**Table 1 animals-12-03239-t001:** Distribution of the significant genomic regions and candidate genes.

Genomic Regions				N of SNPs	Candidate Genes
Chr	Start	End	Kb		
BTA2	81,899,298	82,248,318	349	65	None
	82,355,133	82,378,221	23	7	None
BTA6	66,234,825	66,436,164	201	57	*ATP10D*, *CORIN*
	81,161,826	81,281,680	120	22	*EPHA5*
BTA9	46,061,002	46,211,676	151	38	None
	46,270,931	46,319,445	49	8	None
BTA11	14,444,882	14,944,596	500	89	*MEMO1*, *DPY30*, *SPAST*, *SLC30A6*, *NLRC4*, *YIPF4*
BTA15	11,474,883	12,369,497	895	152	None
BTA16	40,264,305	40,823,586	559	92	*TNFSF18*, *TNFSF4*
	43,872,679	44,009,452	137	18	*TMEM201*, *SLC25A33*
BTA17	15,394,915	15,766,365	371	64	*INPP4B*
BTA21	2,945,936	3,556,313	610	136	None
	4,051,182	4,754,672	703	129	*SCAPER*, *RCN2*, *PSTPIP1*, *TSPAN3*, *PEAK1*
	31,947,095	32,283,141	336	70	*PEAK1*
	32,283,208	32,346,053	63	15	*PEAK1*, *HMG20A*, *LINGO1*
	32,490,146	32,802,506	312	89	None
	36,931,645	37,537,467	606	150	*GABRB3*, *GABRA5*
BTA27	32,891,817	33,308,815	417	56	*ZNF703*, *ERLIN2*, *PLPBP*, *ADGRA2*, *BRF2*, *RAB11FIP1*, *GOT1L1*, *ADRB3*, *EIF4EBP1*, *ASH2L*
	33,448,422	33,466,272	18	6	*NSD3*, bta-mir-2400

## Data Availability

All data supporting this study are included in the article and in the Appendix A.

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
