# Peer review of "Runs of Homozygosity Analysis Reveals Genomic Diversity and Population Structure of an Indigenous Cattle Breed in Southwest China"

_animals, 2022, doi:10.3390/ani12233239_

Round 1
Reviewer 1 Report
Wang and coauthors carried out runs of homozygosity analysis in an indigenous cattle breed, Xieka cattle, which is distributed in Southwest China.
Comments to the Authors:
Authors mentioned that China had many officially recognized and unregistered breeds, so how Xieka can be recognized as an independent breed?
Please include versions for all used software.
What software is used for “The raw sequencing reads in FASTQ format were subjected to quality filtering to discard low-quality reads…”
“reads containing adaptor sequences”, did you remove all reads if they had adopters?
“Against the reference genome, average mapping rate was 99.5% for these clean reads; and by which the 96.9% and 57.2% of genome sequences were covered by at least 1X and 4X sequencing reads, respectively (Table S2).” This indicates that your sequencing depth is extremely low. You need to validate genotyping accuracy using other approaches, such Taqman assay.
“A total of 48,699,312 raw SNPs were initially obtained, which resulted into 796,828 clean SNPs after being subjected to our custom processing steps.” You need to interpret why 98.4% of SNPs are filtered out? If this is true, the number of genotyped SNPs is very similar to the BovineHD DNA SNP chip (N= 777,000 SNPs). You need to further explain why low-depth sequencing is better than SNP chip?
In Figure 1A, what is y-axis? Do you plot SNP individually or plot SNPs based on window?
In Figure 1B, you need provide full name of abbreviations. Do you filter SNPs based on MAF?
“the nucleotide diversity ranged from 0.09 to 0.51 (with the mean of 0.28).” This is really high. Generally the magnitude of nucleotide diversity is at 10-3? How you interpret this extremely high value if this is not mistake?
In Figure 1C, what metric is used for evaluating relatedness among individuals?
In Figure 1D, a circle barplot is better than this one.
“On the whole, there were 3226 ROH with length of 0.5-1 Mb (74%), 970 ROH of 1-2 Mb (22%), 176 ROH of 2-5 Mb (4%), and five ROH >5 Mb, respectively”. How is this compared to other studies?
You mentioned “Significant Genomic Regions”? How do you define this, and what P value threshold do you use?
Do you observe ROH differences between male and female?
How are ROHs in Xieka cattle compared to those in other cattle breeds, even or wild counterparts?
Reviewer 2 Report
A pdf attached with detailed comments/suggestions is provided. I appreciate the application of this manuscript.

Author Response
We really appreciate the extremely careful revisions throughout the whole manuscript, which have significantly improved the quality of this manuscript. We are glad to accept all these revision suggestions, and please see the revised MS for details.
Round 2
Reviewer 1 Report
Comments are below:
“(i) reads containing adaptor sequences”, based on my experience, this is completely wrong. All reads sequenced on Illumina platforms should have adaptors, so here in principle, only adaptors from either end of reads were trimmed off.
“Yes, our samples were subjected to low-depth sequencing of genomes, as being shown by the final sequencing coverages. In the context of studying population genetics based on genome-wide SNPs, we think it is not necessary to validate the candidate SNPs because of the very low error rate for second-generation sequencing techniques. As you mentioned below, only small proportion of raw SNPs were finally retained after QC steps, which would most likely guarantee the accuracy regarding these clean SNPs.” Small proportion of SNPs were remained after qc does not mean they are reliable?
“Because of low-depth sequencing of genomes as discussed above, it is anticipated that a large proportion of raw SNPs will be discarded after our QC steps. The number of clean SNPs (after QC) obtained in this study is similar to that of BovineHD DNA SNP chip, whereas there would be two advantages for using the low-depth genome sequencing. First, the custom SNP chip would be biased on the genotyped SNPs in our case with studying one indigenous breed. Second, we expect that the sequencing data could be used in future for genotype imputation to obtain the genome sequence variants. However, we did not include this information into our manuscript because they are not directly related to the research aims.” If you think SNPs obtained in this study is better suited for this breed-specific analysis, a comparison between detected snps and BovineHD DNA SNP chip should be done, at least their physical positions?
“We obtained a total of 1711 M raw paired sequencing reads”. When the abbreviation “M” first appeared, it should have full name. Throughout the manuscript, writing should follow the scientific rules of the entire community.
“Our analyses confidently revealed that this indigenous cattle breed have remained a relatively high degree of genetic diversity” and “In conclusion, the relatively high degree of genetic diversity of Xieka cattle was revealed using the genomic information; and the proposed candidate genes could help us optimize the breeding programs regarding this indigenous breed”. Based on my evaluation, I do not think the current evidences provide solid supports to draw this conclusion, i.e. Xieka cattle has a high genetic diversity.
Software name for “VCFtool” is different in different sections.
